# The Spectrum of Low-*p*_T_ *J*/*ψ* in Heavy-Ion Collisions in a Statistical Two-Body Fractal Model

**DOI:** 10.3390/e25121655

**Published:** 2023-12-13

**Authors:** Huiqiang Ding, Luan Cheng, Tingting Dai, Enke Wang, Wei-Ning Zhang

**Affiliations:** 1School of Physics, Dalian University of Technology, Dalian 116024, China; dinghuiqiang@mail.dlut.edu.cn (H.D.); dttination@mail.dlut.edu.cn (T.D.); wnzhang@dlut.edu.cn (W.-N.Z.); 2Institute of Quantum Matter, South China Normal University, Guangzhou 510631, China; wangek@scnu.edu.cn

**Keywords:** statistical two-body fractal model, *J*/*ψ* distribution, transverse momentum spectrum

## Abstract

We establish a statistical two-body fractal (STF) model to study the spectrum of J/ψ. J/ψ serves as a reliable probe in heavy-ion collisions. The distribution of J/ψ in hadron gas is influenced by flow, quantum and strong interaction effects. Previous models have predominantly focused on one or two of these effects while neglecting the others, resulting in the inclusion of unconsidered effects in the fitted parameters. Here, we study the issue from a new point of view by analyzing the fact that all three effects induce a self-similarity structure, involving a J/ψ-π two-meson state and a J/ψ, π two-quark state, respectively. We introduce modification factor $q_{TBS}$ and $q_{2}$ into the probability and entropy of charmonium. $q_{TBS}$ denotes the modification of self-similarity on J/ψ, $q_{2}$ denotes that of self-similarity and strong interaction between *c* and $\bar{c}$ on quarks. By solving the probability and entropy equations, we derive the values of $q_{TBS}$ and $q_{2}$ at various collision energies and centralities. Substituting the value of $q_{TBS}$ into distribution function, we successfully obtain the transverse momentum spectrum of low-$p_{T}$ J/ψ, which demonstrates good agreement with experimental data. The STF model can be employed to investigate other mesons and resonance states.

## 1. Introduction

Identified particle spectrum in transverse momenta are pillars in the discoveries of heavy-ion collisions [1,2,3,4,5]. Among the identified particles, J/ψ is produced at the early stage of collisions and interacts with the surroundings during the whole evolution of the system [6,7]. So J/ψ carries significant information and serves as a reliable probe in heavy-ion collisions [8,9].

Charmonium dissociates in quark-gluon plasma (QGP) [10] and can regenerate by a coalescence of *c* and $\bar{c}$ quarks close to the hadronization transition [11]. After the regeneration process, the number of J/ψ is nearly constant [12]. Consequently, the study of the distribution of J/ψ in hadron gas holds significance. Previous models study the process affected by surrounding hadrons from three aspects: (i) the collective flow effect of the expanding hadron gas [13,14], (ii) the quantum correlation effect between J/ψ and neighbouring hadrons [15], (iii) the interaction effect between J/ψ and neighbouring hadrons [16,17]. The typical and representative models are the Tsallis blast-wave (TBW) model [14,18,19,20] and the hadron resonance gas (HRG) model [21,22,23,24]. The TBW model concentrates on aspect (i), the collective flow effect, but ignores aspects (ii) and (iii) [18,19,20]. The authors introduce four parameters to fit RHIC data—temperature *T*, escort parameter *q*, maximum flow velocity $\beta_{s}$, and additional parameter *A* which provides the overall normalization of dN/dy [18,19,20]. All the parameters are determined by fitting experimental data. The earlier HRG model considers aspect (iii)—the interaction effect, but ignores aspects (i) and (ii). By fitting the parameter of the radius of hard core $R_{i}$, the HRG model is used to study the thermal dynamic quantities of hadrons [21]. Hence, previously, the models consider only one aspect and ignore others. The unconsidered effects are taken into the parameters to fit the experimental data [18,19,20,21,22,23,24]. Therefore, it is important to find a method to study the quantities of J/ψ with considering all the effects instead of considering only one aspect; the unconsidered is taken into the fitting data.

In this paper, we study the transverse momentum spectrum of J/ψ from a new point of view. We analyze the fact that the collective flow, quantum and interaction effects all induce J/ψ and its nearest meson to form a J/ψ-π molecule state near to the phase transition critical temperature [25]. From the whole picture of the J/ψ-π molecule state, a two-meson structure can be observed. From the partial picture of the J/ψ meson and the π meson individually, it can be seen that they are both two-quark systems, as shown in Figure 1c. Therefore, in our model, we propose that the J/ψ-π molecule state, as well as J/ψ and π mesons, form a self-similarity structure [26,27] as shown in Figure 1a. With system expansion, the two-meson molecule state and the self-similarity structure disintegrate. We use statistical fractal theory to describe the two-body self-similarity structure. We introduce an influencing factor, $q_{TBS}$, to denote the modification of the two-body self-similarity structure on J/ψ, and an escort factor, $q_{2}$, to denote the modification of self-similarity and binding interaction of heavy quarks on *c* and $\bar{c}$. The preceding models solely account for a single aspect while disregarding others. Unlike the unconsidered effects in those models were taken into the fitting data, we derive the values of $q_{TBS}$ and $q_{2}$ through the solution of probability and entropy equations, taking into consideration the self-similarity structure. Substituting the obtained $q_{TBS}$ into the transverse momentum distribution of J/ψ, we calculate the transverse momentum spectrum and compare results to the experimental data.

## 2. Statistical Two-Body Fractal Model

Near to the critical temperature after regeneration, J/ψ is influenced by the surrounding hadrons from three aspects: the collective flow, quantum correlation and interaction effects. All these effects induce J/ψ and its nearest neighbouring meson to form a J/ψ-π two-hadron structure. This is because

**(1)** in hadron gas, J/ψ co-moves with the nearest neighboring hadron (may well be pion) because of collective flow [13,14];**(2)** the area within a radius of J/ψ’s thermal wavelength accommodates a pion. The wavelength of J/ψ is $\lambda =h/\sqrt{2\pi mkT}$=0.681fm [28]. Near to the critical temperature with T=0.17GeV, the particle number density of pions is $0.5/{\mathrm{fm}}^{3}$ [15], the average distance $l_{a}$ of pions is 1.3 fm. Because $l_{a}<2\lambda$, we can come the the conclusion that within the diameter of 2λ=1.362 fm around J/ψ, a pion has quantum correlation with the J/ψ meson.**(3)** the strong interaction effective distance $l_{Q}$ between quarks is about 0.8 fm [29]. The area within this distance around J/ψ can accommodate a pion, whose particle number density near to the critical temperature is $0.5/{\mathrm{fm}}^{3}$. $l_{a}<2l_{Q}$, so that J/ψ and the nearest neighbouring pion has strong interaction.

Overall, the above analysis shows that the influence of the collective flow, quantum correlation and strong interaction effects induce J/ψ and the nearest neighbouring pion to form a two-body J/ψ-π molecule-state system as shown in Figure 1b. Meanwhile, inside the J/ψ-π molecule-state system, from the quark aspect, J/ψ and π individually are two-quark systems. So in our model, we propose that near to the critical temperature, the J/ψ-π two-meson state from the whole picture, as well as J/ψ and π two-quark systems from the partial picture, satisfy self-similarity [26,27]. Fractal theory has been widely used in investigating systems with self-similarity in different scales [30,31,32]. In recent years, the fractal inspired Tsallis statistical theory is widely used in studying systems with self-similarity fractal structures [33,34,35]. Therefore, here, we use the fractal inspired Tsallis theory to study the self-similarity of J/ψ-π two-body systems. With system expansion, the distance between mesons increases, and most molecule states disintegrate, so the self-similarity structure vanishes.

J/ψ is an energy state of charmonium $c\bar{c}$-bound state. Here, we consider the modification of the two-body self-similarity structure on J/ψ. According to the fractal inspired Tsallis theory, we introduce self-similarity modification factor $q_{TBS}$ to denote modification [33,34,35]. When $q_{TBS}=1$, J/ψ is not modified. The more $q_{TBS}$ deviates from 1, the more J/ψ is modified. In the rest frame, the probability of charmonium at the J/ψ state can be written as [35,36]
(1)$$P_{J/\psi q_{TBS}}=\frac{P1_{1}^{q_{TBS}}}{\sum_{i}P1_{i}^{q_{TBS}}}=\frac{\langle \psi_{1}|{\left[1+\left(q_{TBS}-1\right)\beta \hat{H}\right]}^{\frac{q_{TBS}}{1-q_{TBS}}}|\psi_{1}\rangle }{\sum_{i}\langle \psi_{i}|{\left[1+\left(q_{TBS}-1\right)\beta \hat{H}\right]}^{\frac{q_{TBS}}{1-q_{TBS}}}|\psi_{i}\rangle },$$
where $P1_{1}$ is the probability of charmonium at the J/ψ state without self-similarity. $\psi_{i}$ is the wavefunction of charmonium at different bound states, $\psi_{1}$ corresponds to the J/ψ state. β is the inverse temperature, β=1/T. $\hat{H}$ is the Hamiltonian of the charmonium, $\hat{H}=\frac{{\hat{P}}_{Q1}^{2}}{2m_{Q}}+\frac{{\hat{P}}_{Q2}^{2}}{2m_{Q}}+2m_{Q}+{\hat{V}}_{c\bar{c}}\left(r\right)$, $m_{Q}=1.275$ GeV, *r* is the distance between *c* and $\bar{c}$. $V_{c\bar{c}}\left(r\right)$ is the heavy quark potential [37,38],
(2)$$V_{c\bar{c}}\left(r\right)=-\frac{\alpha_{s}}{r}+\sigma r-\frac{0.8\sigma }{m_{Q}^{2}r},$$
where $\alpha_{s}$ is the strong coupling constant with $\alpha_{s}$ = 0.385 [38], string tension σ = 0.223 ${\mathrm{GeV}}^{2}$ [38]. Here, the mass of the heavy quark is large enough so that the relativistic corrections can be ignored [7]. In many models for calculation convenience, spin effects are neglected [38,39,40]. Here, we also neglect the spin effects; then, the degeneracy factor is set to be 1 for $c,\bar{c}$-bound states.

Partition function $\sum_{i}\langle \psi_{i}|{\left[1+\left(q_{TBS}-1\right)\beta \hat{H}\right]}^{\frac{q_{TBS}}{1-q_{TBS}}}|\psi_{i}\rangle$ is the sum of probabilities over all microstates,
(3)$$\begin{array}{cc} \; & \sum_{i}\langle \psi_{i}|{\left[1+\left(q_{TBS}-1\right)\beta \hat{H}\right]}^{\frac{q_{TBS}}{1-q_{TBS}}}|\psi_{i}\rangle ={\left[1+\left(q_{TBS}-1\right)\beta E_{0}\right]}^{\frac{q_{TBS}}{1-q_{TBS}}}\\ \; & +{\left[1+\left(q_{TBS}-1\right)\beta E_{1}\right]}^{\frac{q_{TBS}}{1-q_{TBS}}}+\ldots +{\left[1+\left(q_{TBS}-1\right)\beta E_{7}\right]}^{\frac{q_{TBS}}{1-q_{TBS}}}\\ \; & +V\int_{|{\vec{p}}_{Q1}|\geq p_{\min}}^{\infty }\int_{|{\vec{p}}_{Q2}|\geq p_{\min}}^{\infty }\int_{r_{\min}}^{r_{\max}}{\left[1+\left(q_{TBS}-1\right)\beta \left(\frac{{\vec{p}}_{Q1}^{2}}{2m_{Q}}+\frac{{\vec{p}}_{Q2}^{2}}{2m_{Q}}+2m_{Q}+V_{c\bar{c}}\left(r\right)\right)\right]}^{\frac{q_{TBS}}{1-q_{TBS}}}\\ \; & \;\;4\pi r^{2}\frac{d^{3}{\vec{p}}_{Q1}d^{3}{\vec{p}}_{Q2}dr}{{\left(2\pi \right)}^{6}}. \end{array}$$

For the lower discrete energy levels, we sum up the eight discrete ones, $\eta_{c}\left(1S\right)$, J/ψ(1S), $h_{c}\left(1P\right)$, $\chi_{c0}\left(1P\right)$, $\chi_{c1}\left(1P\right)$, $\chi_{c2}\left(1P\right)$, $\eta_{c}\left(2S\right)$ and ψ(2S), which are measured in Experiment [41]. $E_{0}$, $E_{1}$, …, $E_{7}$ are the energies of the eight discrete states. For energy levels higher than ψ(2S), the energies are nearly continuous [41]. For convenience of calculation, we integrate the higher energy levels. $p_{\min}$ is the minimum momentum of the higher-level part. Because the difference of the momentum at adjoint energy levels is small [41], we take the momentum of the ψ(2S) state, which is the highest energy level of the eight discrete states, as $p_{\min}$ here. Here, the values of energy levels are obtained by solving the non-relativistic Schrödinger equation [42],
(4)$$\hat{H}\psi_{i}\left(r\right)=E_{i}\psi_{i}\left(r\right),$$
where $\hat{H}={\hat{H}}_{\mathrm{kinetic}}+{\hat{V}}_{c\bar{c}}\left(r\right)=\frac{{\hat{P}}_{Q1}^{2}}{2m_{Q}}+\frac{{\hat{P}}_{Q2}^{2}}{2m_{Q}}+2m_{Q}+{\hat{V}}_{c\bar{c}}\left(r\right)$. Here, because we neglect the spin corrections in the heavy quark potential $V_{c\bar{c}}\left(r\right)$, the energy level differences between $\eta_{c}\left(1S\right)$ and J/ψ(1S), $h_{c}\left(1P\right)$, $\chi_{c0}\left(1P\right)$, $\chi_{c1}\left(1P\right)$ and $\chi_{c2}\left(1P\right)$, $\eta_{c}\left(2S\right)$ and ψ(2S) can be neglected [41]. The detailed values of eigenvalues *E* which correspond to the second row in Table 1 are shown below.

In Equation (Equation 3), V is the volume of charmonium’s motion relative to the surrounding particles with a radius of $r_{0}$. Here, we consider the charmonium in a rest frame, so the volume of charmonium’s motion equals the sum of the motion volume of J/ψ’s neighboring meson (may well be pion) and the volume occupied by J/ψ and the neighboring pion. Therefore, we can write
(5)$$r_{0}=\left(v\tau +d_{J/\psi }+d_{\pi }\right)/2,$$
where *v* is the mean velocity of the surrounding mesons relative to J/ψ. It is dependent on the collision energy and centrality. τ is the lifetime of J/ψ in the medium, $\tau =1/\Gamma \approx 1/0.033\;{\mathrm{GeV}}^{-1}$ [43], $d_{J/\psi }$ and $d_{\pi }$ are the diameters of J/ψ and pion with $d_{J/\psi }+d_{\pi }\approx 2.1\;\mathrm{fm}$ [29].

The values of *v* in Equation (Equation 5) are obtained from the average transverse momentum of J/ψ and π with $v=\sqrt{\left[{\left(p_{TJ/\psi }p_{T\pi }\right)}^{2}-m_{J/\psi }^{2}m_{\pi }^{2}\right]/{\left(p_{TJ/\psi }p_{T\pi }\right)}^{2}}$, where $p_{TJ/\psi }$ and $p_{T\pi }$ are the corresponding momentum within the range of error which comes from experimental data [44,45] or AMPT simulation [46], $m_{J/\psi }=3.096$ GeV, $m_{\pi }=$ 0.139 GeV. For $\sqrt{s_{\mathrm{NN}}}$ = 39GeV, v=0.831,0.809,0.811 (natural unit) for 0–20%, 20–40%, 0–60% centrality, respectively. For $\sqrt{s_{\mathrm{NN}}}=62.4\;\mathrm{GeV}$, v=0.859,0.839,0.841 for 0–20%, 20–40%, 0–60% centrality. For $\sqrt{s_{\mathrm{NN}}}=200\;\mathrm{GeV}$, v=0.884,0.865,0.867 for 0–20%, 20–40%, 0–60% centrality. Substituting the values of *v* into Equation (Equation 5), the values of $r_{0}$ within error range at different collision energies $\sqrt{s_{\mathrm{NN}}}$ and centrality classes are obtained, which is shown in Table 2. It can be seen that with higher collision energies or with more central collisions, $r_{0}$ is larger.

In Equation (Equation 3), $r_{\min}$ and $r_{\max}$ are the lower and upper limits of the distance between *c* and $\bar{c}$. We take the value of the diameter of motion volume $2r_{0}$ as $r_{\max}$, and the minimal spacing 0.05 fm in reference [47] as $r_{\min}$.

In the above, we discussed the escort probability of the charmonium at the J/ψ state with considering the influence of self-similarity structure. Entropy is also an important quantity to study physical properties. So here, we try to analyze the properties of charmonium through self-similarity-influenced entropy. The interaction force we consider here is a strong interaction force. The strong interaction potential is proportional to $r^{-\alpha }$ with α=1 in the weak coupling region, and α=−1 in the strong coupling region [38]. As reference [34] defines, for interaction potential $V\left(r\right)\approx r^{-\alpha }$, if α/d≤1 (*d* is the dimension of the system; here, we consider d=3), the interaction is a long-range interaction. So according to the form of the strong interaction here, regardless of whether it is strongly coupled or weakly coupled, it is a long-range interaction. Tsallis entropy is proved to describe long-range interaction system very well [34,35] and widely used in high-energy physics [48,49]. Meanwhile, Tsallis entropy is related to the escort probability in multifractal [35,50,51,52] and obeys maximum entropy principle. Therefore, here, we use the fractal inspired Tsallis entropy to describe the charmonium system,
(6)$$\begin{array}{cc} S_{J/\psi q_{TBS}}= & \frac{1-\sum_{i}P1_{i}^{q_{TBS}}}{q_{TBS}-1}\\ = & \left(1-\frac{\sum_{i}\langle \psi_{i}|{\left[1+\left(q_{TBS}-1\right)\beta \hat{H}\right]}^{\frac{q_{TBS}}{1-q_{TBS}}}|\psi_{i}\rangle }{\{\sum_{i}\langle \psi_{i}|{\left[1+\left(q_{TBS}-1\right)\beta \hat{H}\right]}^{\frac{1}{1-q_{TBS}}}|\psi_{i}{\rangle \}}^{q_{TBS}}}\right)/\left(q_{TBS}-1\right). \end{array}$$

The above analysis is carried out from the charmonium aspect in the whole picture of the J/ψ-π two-meson state. We propose in our model that the J/ψ-π molecule state and the J/ψ and π mesons form a self-similarity structure. So inside J/ψ, from the quark aspect, as shown in Figure 1a, the probability of the *c* quark and the $\bar{c}$ antiquark also obeys the power-law form. It can be written as [35,36]
(7)$$P_{c}=P_{\bar{c}}=\frac{\langle \phi_{Q1}|{\left[1+\left(q_{Q}-1\right)\beta {\hat{H}}_{Q}\right]}^{q_{Q}/\left(1-q_{Q}\right)}|\phi_{Q1}\rangle }{\sum_{i}\langle \phi_{Qi}|{\left[1+\left(q_{Q}-1\right)\beta {\hat{H}}_{Q}\right]}^{q_{Q}/\left(1-q_{Q}\right)}|\phi_{Qi}\rangle },$$
where $\phi_{Qi}$ is the wavefunction of the heavy quark, $\phi_{Q1}$ corresponds to the wavefunction of the *c* quark when the charmonium is at the J/ψ state. ${\hat{H}}_{Q}$ is its Hamiltonian, ${\hat{H}}_{Q}=\frac{{\hat{p}}_{Q}^{2}}{2m_{Q}}+m_{Q}$, $q_{Q}$ denotes the modification of the self-similarity on the *c* quark, which comes from influence of the strong interaction between *c* and $\bar{c}$ inside J/ψ, and the influence of outside hadrons on J/ψ. The probability of the charmonium at the J/ψ state is the product of the probability of *c* and $\bar{c}$. So we write the probability of charmonium at the J/ψ state as
(8)$$P_{J/\psi q_{2}}=P_{c}\cdot P_{\bar{c}}=\frac{\langle \phi_{1}|{\left[1+\left(q_{2}-1\right)\beta {\hat{H}}_{0}\right]}^{q_{2}/\left(1-q_{2}\right)}|\phi_{1}\rangle }{\sum_{i}\langle \phi_{i}|{\left[1+\left(q_{2}-1\right)\beta {\hat{H}}_{0}\right]}^{q_{2}/\left(1-q_{2}\right)}|\phi_{i}\rangle },$$
where $\phi_{i}$ is the wavefunction of the two-quark system, $\phi_{1}$ corresponds to the state with kinetic energy equal to J/ψ. Here, we define $q_{2}$ to obey equation
(9)$${\left[1+\left(q_{2}-1\right)\beta {\hat{H}}_{0}\right]}^{\frac{q_{2}}{1-q_{2}}}={\left[1+\left(q_{Q}-1\right)\beta {\hat{H}}_{Q}\right]}^{\frac{q_{Q}}{1-q_{Q}}}\cdot {\left[1+\left(q_{\bar{Q}}-1\right)\beta {\hat{H}}_{\bar{Q}}\right]}^{\frac{q_{\bar{Q}}}{1-q_{\bar{Q}}}},$$
where ${\hat{H}}_{0}={\hat{H}}_{Q}+{\hat{H}}_{\bar{Q}}=\frac{{\hat{p}}_{Q}^{2}}{2m_{Q}}+\frac{{\hat{p}}_{{\bar{Q}}^{2}}}{2m_{\bar{Q}}}+2m_{Q}$. In the range of $1<q_{Q}<2$ and the eigen energy larger than the ground state of $c\bar{c}$, we prove by numerical analysis that $q_{2}$ is solvable in Equation (Equation 9), so that it is logical to write Equation (Equation 8) in this form.

Partition function $\sum_{i}\langle \phi_{i}|{\left[1+\left(q_{2}-1\right)\beta {\hat{H}}_{0}\right]}^{q_{2}/\left(1-q_{2}\right)}|\phi_{i}\rangle$ in Equation (Equation 8) is the sum of probabilities of the two-quark system of all the microstates. Similarly to the previous case, we integrate the higher energy levels and sum up the eight discrete lower energy levels. The partition function can be written as
(10)$$\begin{array}{cc} \sum_{i}\langle \phi_{i}|[1+ & \left(q_{2}-1\right)\beta {\hat{H}}_{0}{]}^{q_{2}/\left(1-q_{2}\right)}|\phi_{i}\rangle ={\left[1+\left(q_{2}-1\right)\beta E_{k0}\right]}^{q_{2}/\left(1-q_{2}\right)}\\ \; & +{\left[1+\left(q_{2}-1\right)\beta E_{k1}\right]}^{q_{2}/\left(1-q_{2}\right)}+\ldots +{\left[1+\left(q_{2}-1\right)\beta E_{k7}\right]}^{q_{2}/\left(1-q_{2}\right)}\\ \; & +V_{1}^{2}\int_{|{\vec{p}}_{Q1}|\geq p_{\min}}^{\infty }\int_{|{\vec{p}}_{Q2}|\geq p_{\min}}^{\infty }{\left[1+\left(q_{2}-1\right)\beta \left(\frac{{\vec{p}}_{Q1}^{2}}{2m_{Q}}+\frac{{\vec{p}}_{Q2}^{2}}{2m_{Q}}+2m_{Q}\right)\right]}^{q_{2}/\left(1-q_{2}\right)}\\ \; & \;\;\frac{d^{3}{\vec{p}}_{Q1}d^{3}{\vec{p}}_{Q2}}{{\left(2\pi \right)}^{6}}, \end{array}$$
where $V_{1}$ is the motion volume of *c* and $\bar{c}$. Here, we take an approximation that the motion volume of *c* and $\bar{c}$ is approximately equivalent to the motion volume of J/ψ, $V_{1}=V$. Also, $E_{k0}$, $E_{k1},\ldots ,E_{k7}$ are the kinetic energies of *c* and $\bar{c}$ at the eight discrete states. They are obtained from the Schrödinger Equation (Equation 4), the detailed values of $E_{k1},\ldots ,E_{k7}$ are shown in the third row of Table 1.

For long-ranged interactions, similar to Equation (Equation 6), the Tsallis entropy of the charmonium can be written as [35,50,51,52]
(11)$$S_{J/\psi q_{2}}=\left(1-\frac{\sum_{i}\langle \phi_{i}|{\left[1+\left(q_{2}-1\right)\beta {\hat{H}}_{0}\right]}^{\frac{q_{2}}{1-q_{2}}}|\phi_{i}\rangle }{\{\sum_{i}\langle \phi_{i}|{\left[1+\left(q_{2}-1\right)\beta {\hat{H}}_{0}\right]}^{\frac{1}{1-q_{2}}}|\phi_{i}{\rangle \}}^{q_{2}}}\right)/\left(q_{2}-1\right).$$

Overall, we analyze the charmonium from meson and quark aspects. From the meson aspect in the whole picture, we consider that the J/ψ meson satisfies self-similarity. We introduce modification factor $q_{TBS}$ and obtain the probability of charmonium $P_{J/\psi q_{TBS}}$ in Equation (Equation 1) and entropy $S_{J/\psi q_{TBS}}$ in Equation (Equation 6). From the quark aspect in the partial picture as shown in Figure 1c, the *c* and $\bar{c}$ quarks also satisfy self-similarity. The probability of *c* and $\bar{c}$ quarks obeys the power-law form. The probability of the charmonium is the product of that of *c* and $\bar{c}$ quarks. We introduce escort parameter $q_{2}$ and obtain the probability of charmonium $P_{J/\psi q_{2}}$ in Equation (Equation 8), entropy $S_{J/\psi q_{2}}$ in Equation (Equation 11). Regardless of aspect, the properties of the charmonium are unchanged; we have
(12)$$P_{J/\psi q_{TBS}}=P_{J/\psi q_{2}};$$
(13)$$S_{J/\psi q_{TBS}}=S_{J/\psi q_{2}}.$$

By placing the different values of $r_{0}$ within the range of error in Table 2 into Equations (Equation 1), (Equation 6), (Equation 8) and (Equation 11), we solve the conservation equations of probability and entropy Equations (Equation 12) and (Equation 13), and obtain the values of $q_{TBS}$ and $q_{2}$ within the error range at different collision energies and centrality classes as shown in Table 3.

Here, $q_{TBS}$ denotes the modification of self-similarity structure on J/ψ; it is an important physical quantity to study the self-similarity influence. We also study the evolution of $q_{TBS}$ with the temperature near to the critical temperature. Shown in Figure 2 is influencing factor $q_{TBS}$ at different fixed temperatures with $r_{0}=$ 3.48, 3.56, 3.63 fm, which is the radius of motion volume of the charmonium relative to surrounding particles at $\sqrt{s_{\mathrm{NN}}}$ = 39, 62.4, 200 GeV for 0–20% centrality which is shown in Table 2. It is found that $q_{TBS}$ is larger than 1. This comes from the value for Tsallis entropy, $S_{q}<S_{B-G}$ if q>1 [35]. Here, the self-similarity structure decreases the number of microstates. So the entropy is decreased and the value of $q_{TBS}$ is larger than 1. At fixed $r_{0}$, the value of $q_{TBS}$ decreases with decreasing the temperature. This is consistent with the fact that J/ψ is typically influenced near to the critical temperature. With system expansion and temperature decreasing, the influence decreases. So $q_{TBS}$ decreases to approaching 1. It is also found that at fixed temperature, influencing factor $q_{TBS}$ increases with increasing $r_{0}$. This is because in a larger motion volume, the probability of the charmonium being influenced by the surroundings is larger, so that influencing factor $q_{TBS}$ is larger.

## 3. Transverse Momentum Spectrum

In the previous section, we established the STF model and derived influencing factor $q_{TBS}$ for J/ψ at various collision energies. In this section, based on influencing factor $q_{TBS}$, we calculate the transverse momentum distribution of J/ψ.

Now, we consider the charmonium as a grand canonical ensemble. Based on the probability in Equation (Equation 1), the normalized density operator $\hat{\rho }$ is [53,54,55,56]
(14)$$\hat{\rho }=\frac{{\left[1+\left(q_{TBS}-1\right)\beta \left(\hat{H}-\mu \hat{N}\right)\right]}^{q_{TBS}/\left(1-q_{TBS}\right)}}{\mathrm{Tr}\;{\left[1+\left(q_{TBS}-1\right)\beta \left(\hat{H}-\mu \hat{N}\right)\right]}^{q_{TBS}/\left(1-q_{TBS}\right)}},$$
where μ is the chemical potential, $\hat{N}$ is the particle number operator of the grand canonical ensemble. We consider that the particles at different microstates, such as $\eta_{c}\left(1S\right)$, J/ψ(1S), and $h_{c}\left(1P\right)\ldots$, to be subsystems of the charmonium system, respectively. We accept the factorization hypothesis that for a system containing subsystems, the thermal system obeys the pseudoadditivity law [53,54,55] as
(15)$$\ln\left[1+\left(q_{TBS}-1\right)\beta \left(E-\mu N\right)\right]=\sum_{i=0}\ln\left[1+\left(q_{TBS}-1\right)\beta \left(\epsilon_{i}-\mu n_{i}\right)\right],$$
where $\epsilon_{i},n_{i}$ are the energy and particle number of each subsystem.

With the density operator, the average particle number of subsystem *i* can be written as [53,54,55]
(16)$${\bar{n}}_{i}=\mathrm{Tr}\;\hat{\rho }\;n_{i}=\frac{1}{{\left[1+\left(q_{TBS}-1\right)\beta \left(\epsilon_{i}-\mu \right)\right]}^{q_{TBS}/\left(q_{TBS}-1\right)}-1},$$
so that the particle number distribution of J/ψ is ${\bar{n}}_{J/\psi }=\frac{1}{{\left[1+\left(q_{TBS}-1\right)\beta \left(\epsilon_{J/\psi }-\mu \right)\right]}^{q_{TBS}/\left(q_{TBS}-1\right)}-1}$.

With the above particle number distribution of J/ψ, the transverse momentum distribution in terms of rapidity *y* can be obtained [53,57],
(17)$$\frac{d^{2}N}{2\pi p_{T}dp_{T}dy}=gV_{lab}\frac{m_{T}\cos hy}{{\left(2\pi \right)}^{3}}{\left\{{\left[1+\left(q_{TBS}-1\right)\beta \left(m_{T}\cos hy-\mu \right)\right]}^{q_{TBS}/\left(q_{TBS}-1\right)}-1\right\}}^{-1},$$
where $m_{T}$ is the transverse mass of J/ψ with $m_{T}=\sqrt{m^{2}+p_{T}^{2}}$, *m* is the mass of J/ψ with m=3.096 GeV, $p_{T}$ is the transverse momentum in the lab frame. β is the inverse of temperature, β=1/T, with T=0.17 GeV. We set the degeneracy factor *g* to be one because the spin effects are ignored. Chemical potential μ is approximately 0 for J/ψ in high energy physics. $V_{lab}$ is J/ψ’s motion volume in the lab frame with $V_{lab}=\gamma V$, *V* is the motion volume in the center of the mass frame of J/ψ in Equation (Equation 3), γ is the Lorentz factor.

In heavy-ion collisions at RHIC energies, the mean number of produced $c\bar{c}$ pairs $N_{c\bar{c}}$ is approximately 1.0 in the central rapidity region [58,59]. Therefore, the transverse momentum distribution of J/ψ in mid-rapidity in hadron gas is approximately the transverse momentum distribution of J/ψ in its motion volume. By substituting the value of obtained $q_{TBS}$ which is shown in Table 3 into transverse momentum distribution of J/ψ in Equation (Equation 17), the transverse momentum spectrum of low-$p_{T}$J/ψ can be obtained.

Shown in Figure 3, Figure 4 and Figure 5 are the transverse momentum spectra of low-$p_{T}$J/ψ for Au–Au collisions at $\sqrt{s_{\mathrm{NN}}}=$ 39, 62.4, 200 GeV and 0–20%, 20–40%, 0–60% centrality classes. We compare our theoretical results with the experimental data [60,61] at a low-$p_{T}$ region. Our theoretical results show a good agreement with the experimental data.

## 4. Conclusions

We establish a statistical two-body fractal (STF) model to study the low-$p_{T}$ transverse momentum spectrum of J/ψ in heavy-ion collisions. After the regeneration process, the number of J/ψ is nearly constant. The distribution of J/ψ in hadron gas is influenced by flow, quantum and strong interaction effects. We comprehensively examine all three effects simultaneously from a novel fractal perspective based on the STF model through model calculation rather than relying solely on data fitting. Close to the critical temperature, the combined action of the three effects leads to the formation of a two-meson structure with its nearest neighboring meson. With the evolution of the system, most of these states undergo disintegration. To describe this physical process, our model proposes that under the influence of the three effects near to the critical temperature, a self-similarity structure emerges, involving a J/ψ-π two-meson state and a J/ψ, π two-quark state, respectively. As the system evolves, the two-meson structure gradually disintegrates. We introduce influencing factor $q_{TBS}$ to denote the modification of two-body self-similarity structure on J/ψ and escort factor $q_{2}$ to denote the modification of self-similarity and binding interaction between *c* and $\bar{c}$. By solving the probability and entropy equations, we derive the values of $q_{TBS}$ and $q_{2}$ at various collision energies and centrality classes. We also analyze the evolution of $q_{TBS}$ with temperature. Interestingly, we observe that $q_{TBS}$ is greater than one and decreases as the temperature decreases. This behavior arises from the fact that the self-similarity structure reduces the number of microstates, leading to $q_{TBS}>1$. The decrease in $q_{TBS}$ with system evolution aligns with the understanding that self-similarity diminishes as the system expands. Substituting the values of $q_{TBS}$ into the distribution function, we successfully obtain the transverse momentum spectrum of low-$p_{T}$J/ψ, which demonstrates good agreement with experimental data. In the future, the STF model can be employed to investigate other mesons and resonance states.

## Figures and Tables

**Figure 1 entropy-25-01655-f001:**
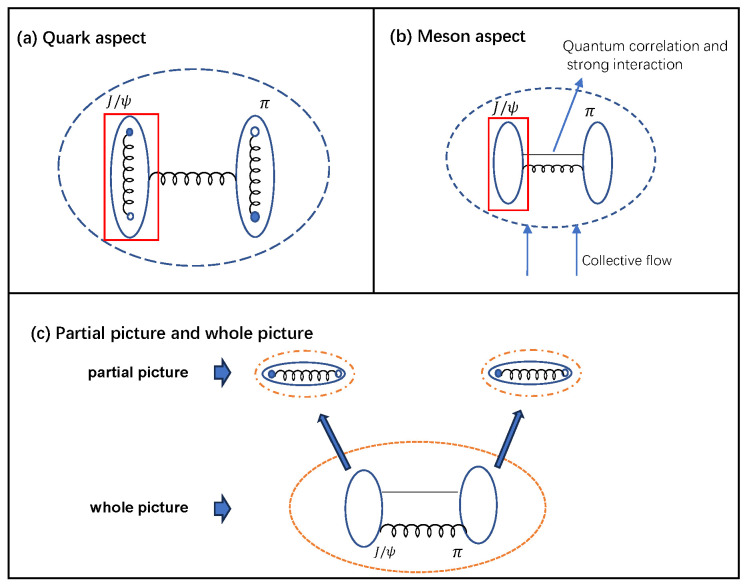
The self-similarity structure of *c* and $\bar{c}$ in the hadron gas near to the critical temperature. (**a**) J/ψ in hadron gas from the quark aspect; (**b**) J/ψ in hadron gas from the meson aspect; (**c**) J/ψ-π two-body self-similarity structure from the partial picture and the whole picture.

**Figure 2 entropy-25-01655-f002:**
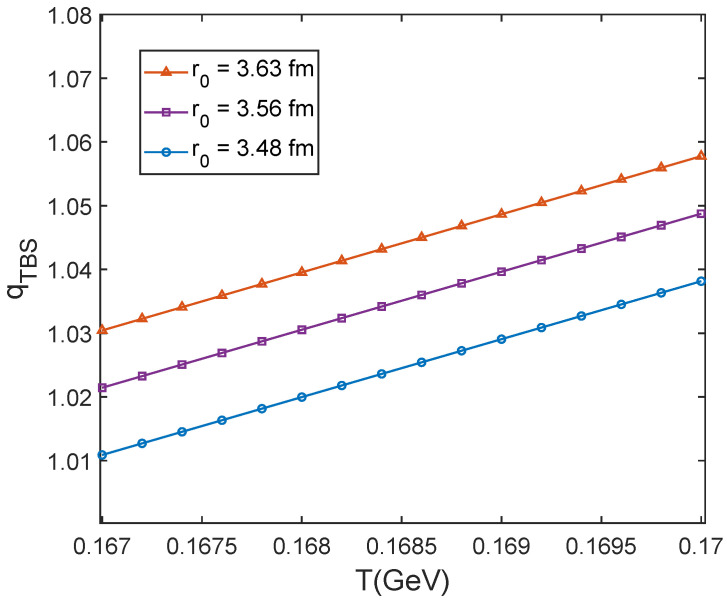
Influencing factor $q_{TBS}$ at different fixed temperature swith $r_{0}=3.48,3.56,3.63\;\mathrm{fm}$.

**Figure 3 entropy-25-01655-f003:**
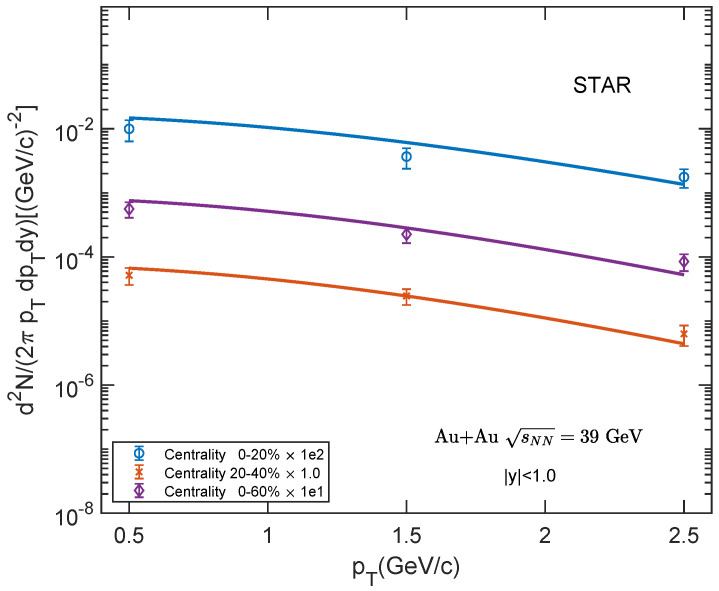
Transverse momentum spectra of J/ψ in Au–Au collisions at $\sqrt{s_{\mathrm{NN}}}$ = 39 GeV for different centrality classes in mid-rapidity region |y|<1.0. The experimental data are taken from STAR [60].

**Figure 4 entropy-25-01655-f004:**
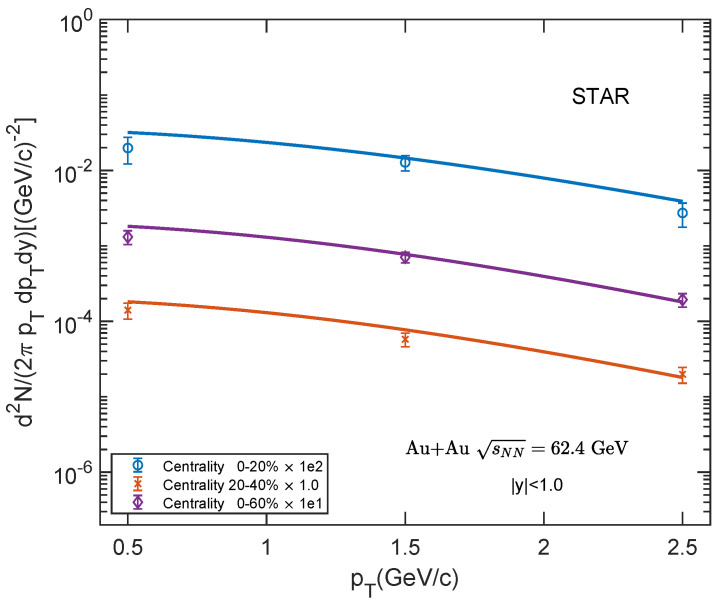
Transverse momentum spectra of J/ψ in Au–Au collisions at $\sqrt{s_{\mathrm{NN}}}$ = 62.4 GeV for different centrality classes in mid-rapidity region |y|<1.0. The experimental data are taken from STAR [60].

**Figure 5 entropy-25-01655-f005:**
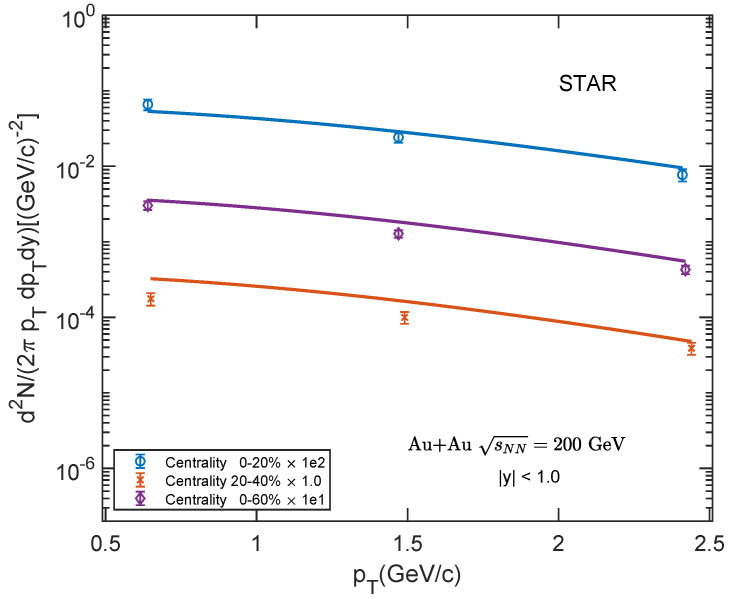
Transverse momentum spectra of J/ψ in Au–Au collisions at $\sqrt{s_{\mathrm{NN}}}$ = 200 GeV for different centrality classes in mid-rapidity region |y|<1.0. The experimental data are taken from STAR [61].

**Table 1 entropy-25-01655-t001:** In the first row, $m_{exp}$ is the rest mass of the eight discrete charmonium states measured in experiments in [41]. The second and third row are energy eigenvalues *E* and kinetic energies $E_{k}$ of the eight discrete states that were obtained by solving the Schrödinger equation in Equation (Equation 4) with neglecting spin corrections, respectively.

State	$\eta_{c}\left(1S\right)$	J/ψ(1S)	$h_{c}\left(1P\right)$	$\chi_{c0}\left(1P\right)$	$\chi_{c1}\left(1P\right)$	$\chi_{c2}\left(1P\right)$	$\eta_{c}\left(2S\right)$	ψ(2S)
$m_{exp}$(GeV)	2.983	3.096	3.525	3.414	3.510	3.556	3.637	3.686
*E*(GeV)	3.047	3.047	3.517	3.517	3.517	3.517	3.792	3.792
$E_{k}$(GeV)	2.926	2.926	2.993	2.993	2.993	2.993	3.071	3.071

**Table 2 entropy-25-01655-t002:** The values of $r_{0}$ within error range at different collision energies $\sqrt{s_{\mathrm{NN}}}=$ 39 GeV, 62.4 GeV, 200 GeV of Au-Au collisions in 0–20%, 20–40%, 0–60% centrality classes.

Au-Au $\sqrt{s_{\mathrm{NN}}}$	$r_{0}$ (fm)		
	0–20% Centrality	20–40% Centrality	0–60% Centrality
39 GeV	3.48 ± 0.04	3.41 ± 0.05	3.42 ± 0.05
62.4 GeV	3.56 ± 0.06	3.50 ± 0.07	3.50 ± 0.07
200 GeV	3.63 ± 0.08	3.57 ± 0.08	3.58 ± 0.09

**Table 3 entropy-25-01655-t003:** Influencing factors $q_{TBS}$ and $q_{2}$ within range error in Au–Au collisions at $\sqrt{s_{\mathrm{NN}}}=$39GeV, 62.4GeV, 200GeV in mid-rapidity region |y|<1.0 for different centrality classes.

Au-Au $\sqrt{s_{\mathrm{NN}}}$		Centrality		
		0–20%	20–40%	0–60%
39 GeV	$q_{TBS}$	1.0381±0.0061	1.0286±0.0068	1.0300±0.0069
	$q_{2}$	1.5861±0.0022	1.5895±0.0024	1.5890±0.0024
62.4 GeV	$q_{TBS}$	1.0487±0.0092	1.0408±0.0107	1.0408±0.0107
	$q_{2}$	1.5826±0.0035	1.5851±0.0039	1.5851±0.0039
200 GeV	$q_{TBS}$	1.0577±0.0096	1.0500±0.0119	1.0513±0.0132
	$q_{2}$	1.5786±0.0045	1.5816±0.0045	1.5811±0.0050

## Data Availability

Data is contained within the article.

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
