# Peer review of "The Spectrum of Low-pT J/ψ in Heavy-Ion Collisions in a Statistical Two-Body Fractal Model"

_entropy, 2023, doi:10.3390/e25121655_

Round 1
Reviewer 1 Report (Previous Reviewer 3)
Comments and Suggestions for Authors
The authors have addressed most part of my concerns from the previous review, and in my opinion the manuscript is acceptable for publication in its present form. I appreciate the careful consideration the authors put into addressing the concerns of myself and the other reviewers.
Comments on the Quality of English LanguageMinor editing of English language required.
Author Response
Please see the attachment

Reviewer 2 Report (Previous Reviewer 2)
Comments and Suggestions for Authors
Referee report on "The Spectrum of Low-$p_{T}$ $J/\Psi$ in Heavy-ion Collisions in a Statistical Two-body Fractal Model", by Huiqiang Ding, Luan Cheng, Tingting Dai, Enke Wang and Wei-Ning Zhang, submitted to Entropy.
The present paper studies the spectra of $J/\Psi$ on the basis of the transverse momentum distribution of the Tsallis-3 statistics in the factorization approximation of the zeroth term approximation (Eq.~(17)) and the so called a Statistical Two-body Fractal Model. The paper is well written and presents some interest. However, both models presented in this paper, which are based on the Tsallis statistics, are theoretically inconsistent. Therefore, I cannot recommend the publication of this paper in Entropy. Nevertheless, below there are a few comments or questions that I would suggest the author considering.
1.) In the single-hadron partition functions (3) and (10), there are inexactitudes. The authors should explain why they use the classical Maxwell-Boltzmann statistics of $c$ and $\bar{c}$ quarks instead of the Fermi-Dirac statistics. The Boltzmann factor in the continuous spectrum of the partition functions (3) and (10) was lost. See the textbooks on statistical mechanics.
2.) The authors should show in more detail the calculations of the eigenvalues and eigenstates (wave functions) of the Schrodinger equation (4) with interaction and without interaction.
3.) In equations (12) and (13), the authors suppose that the system of $c$ and $\bar{c}$ quarks with the strong interaction is the same as the system of this quark-antiquark pair without interaction. Such a hypothesis for the probability and the entropy is not founded. However, if the authors consider the phase transition then they should write the Gibbs conditions for the corresponding statistical ensemble.
4.) The probability (1) corresponds to the thermodynamic system of one hadron in different internal states. However, the transverse momentum distribution (17) corresponds to the thermodynamic system of many hadrons without internal states. Thus, they are not compatible. The authors erroneously consider that the volume in the transverse momentum distribution (17) is the volume of one $J/\psi$ hadron.
5.) Probability (1) does not correspond to the Tsallis-1, Tsallis-2, or Tsallis-3 statistics. This probability corresponds in form to the Tsallis-3 statistics. However, in the Tsallis-3 statistics, the microstate probability contains two normalization functions missing in equation (1). These normalization functions are the solutions of a system of two normalization equations, which are absent in this work. Thus, the probability (1) is inconsistent. The authors should carefully formulate their model in terms of the Tsallis-3 statistics.
6.) Equation (6) contradicts Eq. (1). In Eq. (6), the microstate probability of the Tsallis-2 statistics was used. However, in Eq. (1), the incorrect probability of the Tsallis-3 statistics was given. Moreover, the Tsallis-2 statistics is generally inconsistent due to the erroneous definition of the generalized mean values, $\langle 1\rangle \neq 1$.
7.) The proof of Eq. (8) is superfluous. It introduces two additional unused parameters, $q_{Q}$ and $q_{\bar{Q}}$. This proof is not convincing and leads to confusion because the probabilities in the Tsallis statistics do not factorize. The authors should prove that Eq. (9) has solutions. Otherwise, Eq. (8) for non-interacting quarks may be introduced by definition.
8.) Equation (11) contradicts Eq. (8) (see the point 6.) given above).
9.) The transverse momentum distribution (17) for the Fermi-Dirac statistics of particles is the transverse momentum distribution of the Tsallis-3 statistics in the factorization approximation of the zeroth term approximation (see the paper [Parvan, J. Phys. G: Nucl. Part. Phys. 50 (2023) 125002]). This distribution is mathematically inconsistent because it corresponds to the factorization approximation. In the Tsallis statistics, the factorization approximation is mathematically invalid as it factorizes a power-law function as if it were an exponential one (see Eq. (15)). The authors should use the exact transverse momentum distributions of the Tsallis-3 statistics.

Author Response
Please see the attachment

Reviewer 3 Report (Previous Reviewer 1)
Comments and Suggestions for Authors
The authors addressed the raised questions and comments. I spotted one misrpint and still I would turn their attention to a possible inconsistency that they could address.
Answer to the first author, 9/20:
2*lambda=1.362
The investigated effects area in-medium effects, but the pion number density is at the freeze-out. Isn't that inconsistent? What is the error that approximation causes?
Round 2
Reviewer 2 Report (Previous Reviewer 2)
Comments and Suggestions for Authors
Referee report on "The Spectrum of Low-$p_{T}$ $J/\Psi$ in Heavy-ion Collisions in a Statistical Two-body Fractal Model", by Huiqiang Ding, Luan Cheng, Tingting Dai, Enke Wang and Wei-Ning Zhang, resubmitted to Entropy.
The answers of the authors are unsatisfactory. In this paper, the Tsallis-3 statistics was not implemented correctly. Therefore, I cannot recommend the publication of this paper in Entropy. Below there are a few comments or questions which were not resolved by the authors.
1.) The authors did not resolve the first question. In the continuous spectrum of charmonium the formulae (3) and (10) were written for the classical Maxwell-Boltzmann statistics of quarks. But the quarks are fermions and the formulae (3) and (10) should be written for the Fermi-Dirac statistics of particles (see, e.g., the textbook [K. Huang, Statistical Mechanics, John Wiley and Sons, New York, 1987]). Therefore, it should be explained why the classical Maxwell-Boltzmann statistics of quarks was used instead of the Fermi-Dirac statistics. Why is it possible for such a system?
2.) The second question was not resolved completely. The authors should write the explicit analytical formulae (maybe in Appendix) for the wave functions of the Schrodinger equation (4) for the system with interaction and without interaction.
3.) The explanations given by the authors are confusing. The qualitative picture depicted in Figure 1 is for the hadron gas. However, the equations (12) and (13) are written for the thermodynamical system of only one hadron in different exited states when the constituent quark and antiquark interact (left side of equations (12) and (13) -- interacting gas of two particles) and when the constituent quark and antiquark are free (right side of equations (12) and (13) -- ideal gas of two particles). The authors should also explain in the text of the manuscript why the temperature and the entropy are the same for the interacting gas of two particles and for the ideal gas of two particles.
4.) The answer of the authors is confusing. The derivation of the transverse momentum distribution (17) from Eq. (1) is incorrect as Eq. (15) is mathematically erroneous. In the Tsallis statistics, the factorization approximation (Eq. (15)) is mathematically invalid as it factorizes a power-law function as if it were an exponential one. Thus, the transverse momentum distribution (17) is inconsistent. The authors should state clearly into the text of the manuscript that their model is valid only for one hadron (the motion volume of $J/\Psi$) and it can be applied to describe the experimental data of those experiments in which only one charmed hadron is created. This situation is not physical.
5.) In this paper, the Tsallis-3 statistics was implemented incorrectly. The physical results obtained in terms of the renormalized temperature $\beta$ are incorrect because $\beta$ is not an independent variable of state. In the Tsallis-3 statistics, the temperature $\beta'$ (see notations from the authors response) is an independent variable of state as it is an independent variable of state in the maximum entropy principle. Thus, only $\beta'$ is the temperature and $\beta$ is an auxiliary function. Moreover, the probability depends on two unknown functions $U_{q}$ and $\sum_{i} p_{i}^{q}$ which should be fixed from the two norm equations. After solving these equations the thermodynamic quantities will be functions of the independent variables of state $(\beta',V,N,q)$.
9.) The answer of the authors is confusing. Equation (15) is mathematically invalid (see the point 4.) given above). The acceptance by many authors does not make it mathematically correct. Thus, the transverse momentum distribution (17) is inconsistent and all the obtained results of the authors are questionable from the point of view of the fundamentals of the statistical mechanics.

Author Response
Please see the attached file.

This manuscript is a resubmission of an earlier submission. The following is a list of the peer review reports and author responses from that submission.
Round 1
Reviewer 1 Report
Comments and Suggestions for Authors
However, the basic idea of the paper might be relevant and worth to be investigated, the discussion is very hard to follow and potentially wrong. I attach a PDF with all my corrections and claim for clarification.
There are several points where I suspect misunderstanding and I suggest to carefully reconsider some of the paper's statements. At the present form it cannot be considered to be published.

Comments on the Quality of English LanguageAn extensive editing and careful check, may be from a native speaker, seems to be necessary. I comment here and there on the language.
Reviewer 2 Report
Comments and Suggestions for Authors
Referee report on "The Spectrum of Low-$p_{T}$ $J/\Psi$ in Heavy-ion Collisions in a Statistical Two-body Fractal Model", by Huiqiang Ding, Luan Cheng, Tingting Dai, Enke Wang and Weining Zhang, submitted to Entropy.
The present paper studies the spectra of $J/\Psi$ on the basis of the transverse momentum distribution of the Tsallis-3 statistics in the factorization approximation of the zeroth term approximation (Eq. (13)). In the Tsallis statistics, the factorization approximation is evidently mathematically invalid as it factorizes a power-law function as if it were an exponential one. The paper is written well and it presents some interest. The authors' calculations agree with the experimental data. However, the theoretical basis of this paper is questionable. The proposed model is theoretically inconsistent. It is like an eclecticism. Therefore, I cannot recommend the publication of this paper in Entropy. Nevertheless, below there are a few comments or questions that I would suggest the author considering.
1.) In the single-hadron partition functions (2) and (7), there are inexactitudes. The authors should explain why they use the classical Maxwell-Boltzmann statistics of $c$ and $\bar{c}$ quarks instead of the Fermi-Dirac statistics. Why were the relativistic corrections in the Hamiltonian of the charmonium in the partition functions (2) and (7) neglected? Why were the degeneracy factors in the discrete spectrum of the partition functions (2) and (7) lost? Why was the Boltzmann factor in the continuous spectrum of the partition functions (2) and (7) lost?
2.) In the Tsallis entropies (5) and (10), the authors use the Boltzmann-Gibbs probabilities of microstates (4) and (9) in which the Boltzmann-Gibbs inverse temperature $\beta$ is multiplied by the parameters $q_{fqs}$ and $q_{2}$, respectively. The Boltzmann-Gibbs probabilities of microstates (4) and (9) correspond to the Boltzmann-Gibbs entropy. Thus, the introduction of the Boltzmann-Gibbs probabilities in the Tsallis entropy violates the principle of thermodynamic equilibrium (the principle of maximum entropy), which is the basis of the Boltzmann-Gibbs and Tsallis statistics. Therefore, in the Tsallis entropies (5) and (10), the authors should use the probabilities of microstates of the Tsallis statistics (Tsallis-1 or Tsallis-3) (see [C. Tsallis, R.S. Mendes, A.R. Plastino, Physica A 261 (1998) 534]).
3.) The authors consider that the factors $q_{fqs}$ and $q_{2}$ represent the flow, quantum and strong interaction effects. This is not obvious. Thus, the authors should prove mathematically these relations otherwise remove these statements from the text.
4.) The authors should show in more detail the calculations of the energies of the discrete spectra in equation (7). The numerical values of the energies of the discrete spectra from the equations (2) and (7) should be compared in a Table.
5.) In equations (11) and (12), the authors suppose that the system of $c$ and $\bar{c}$ quarks with the strong interaction is the same as the system of this quark-antiquark pair without interaction. Such a hypothesis is erroneous.
6.) The probability (4) corresponds to the thermodynamic system of one hadron in different internal states. However, the transverse momentum distribution (13) corresponds to the thermodynamic system of many hadrons without internal states. Thus, they are not compatible.
7.) The authors should derive analytically equation (13) from the entropy (5) to show their correspondence.

Reviewer 3 Report
Comments and Suggestions for Authors
Report of the Referee
Manuscript Ref.: Entropy-2596350
Title: "The Spectrum of Low-$p_{T}$ $J/\psi$ in Heavy-ion Collisions in a
Statistical Two-body Fractal Model"
===========================================================================================
The authors propose to describe the low transverse momentum of quarkonium production in AA collisions at RICH energies by using an interesting statistical two-body fractal model. It is claimed that the model effectively includes flow, quantum (coorelations) and strong interaction effects in the context of self-similarity structure (properties of bound state quarkonium+pion and the correspondig quark system are the same). Those effects are absorbed in only two parameters, $q_{fqs}$ and $q_2$, which are theoretically determined via Eqs. (11-12) by using the expressions for probability and entropy presented in Eqs.(4-5) and (9-10), respectively. Moreover, Tsallis entropy is assumed to describe both systems.
The paper is relatively well presented and the references are adequate. The topic is of interest in the community. However, I have some questions about the results and comparison to data.
Different models have been proposed to describe the hadron momentum spectra and those based on the Tsallis distribution produce good description of the spectra in minimum bias or moderate multiplicity proton-proton collisions. The same is true also for the two component models based on the sum of Boltzmann-Gibbs distribution and a power-law term. However, for heavy-ion collisions these models fail to describe the measured spectra in the whole available transverse momentum range and more complex models have been suggested. Very often hydrodynamical flow effects are introduced through the blast-wave model (BWM), which describe the (thermal) low $p_T$ range very well. On the other hand, authors used the Tsallis distribution in Eq. (13). They cited Refs. [6] and [49] but these publications refer to the comparison of Tsallis distritution in pp collisions only. Could the authors discuss and compare their results to the BWM model ? Does the proposed model effectively describe the thermal and collective part of the hadron spectrum? The meaning of the parameters $q_{fqs}$ and $q_2$ has to be elaborated carefully. In my view it is not clear which physical effects they are really introducing (flow, energy loss, nuclear shadowing, particle correlations?). In the expression for $f_i$ after Eq. (13) the $\beta$ parameter is not defined. Is it the inverse of temperature? How the temperature is computed in the present model? What is the value for the parameter V and its meaning in the expression for $V_{lab}$ normalizing $f_i$?
For the reasons presented above the paper is relatively clear and valuable. It is suitable in my view for the Entropy journal after the questions to be addressed.
Comments on the Quality of English Language
Minor editing of English language required.